# Perch 2.0: The Bittern Lesson for Bioacoustics

## Abstract

Perch is a performant pre-trained model for bioacoustics. It was trained in supervised fashion, providing both off-the-shelf classification scores for thousands of vocalizing species as well as strong embeddings for transfer learning. In this new release, Perch 2.0, we expand from training exclusively on avian species to a large multi-taxa dataset. The model is trained with self-distillation using a prototype-learning classifier as well as a new source-prediction training criterion. Perch 2.0 obtains state-of-the-art performance on the BirdSet and BEANS benchmarks. It also outperforms specialized marine models on marine transfer learning tasks, despite having almost no marine training data. We present hypotheses as to why fine-grained species classification is a particularly robust pre-training task for bioacoustics.

## 1 Introduction

Bioacoustics is an important tool in the fields of biology and ecology with vital applications in conservation and biodiversity monitoring (Laiolo, 2010). In recent years machine learning methods such as deep learning have largely replaced (Stowell, 2022) more traditional signal processing methods such as template matching (Towsey et al., 2012) for event detection and classification in bioacoustics. Moreover, models trained on large amounts of avian, labeled bioacoustics data such as BirdNET (Kahl et al., 2021) and the previous iteration of Perch (Perch 1.0, Hamer et al., 2023) have been shown to transfer to novel tasks such as individual identification (Huang et al., 2025) as well as non-avian bioacoustics problems (Ghani et al., 2023).

In this paper we introduce Perch 2.0, a new iteration of the Perch model that achieves state-of-the-art results on two bioacoustics benchmarks: BirdSET and the BEnchmark of ANimal Sounds (BEANS) (Section 3). Perch 2.0 was primarily trained with the task of species classification. Self-supervised learning has been explored in the literature as a way to further improve bioacoustics models—Bird-MAE (Rauch et al., 2025a), BirdAVES (Hagiwara, 2023) and SimCLR-style models (Moummad et al., 2024)— but these approaches have had mixed results, with some works (Ghani et al., 2023; Kather et al., 2025) arguing in favor of supervised models while others Schwinger et al. (2025) have argued in favor of self-supervises models. In our exploratory research we too have experimented with a variety of self-supervised methods such as MAEs, HuBERT and SimCLR but experienced a similar inability to consistently outperform supervised models. With Perch 2.0 we present a demonstration that a simple, well-tuned supervised bioacoustic foundation model can in fact match the

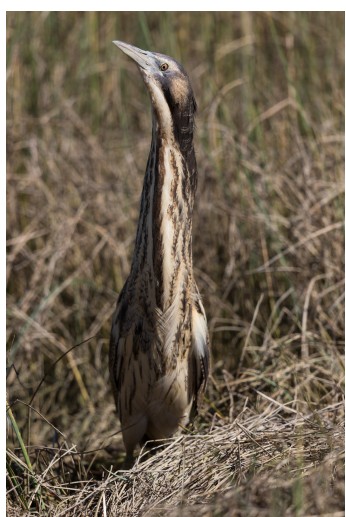

Figure 1: In the famous essay *The Bitter Lesson* (Sutton, 2019) it was argued that advancements in AI come from simple, general-purpose methods. The 'bittern lesson' from Perch 2.0 is that simple, supervised models are difficult to beat. Bitterns are a subfamily of herons; the Australasian bittern pictured here is an elusive, endangered species whose distinctive booming vocalizations can carry long distances. (Image by Imogen Warren licensed under CC BY-SA 4.0.)

performance of or outperform the best self-supervised bioacoustic foundation models. We present some hypotheses as to why the fine-grained task of (avian) species classification is a particularly useful pre-training task (Section 4).

Compared to previous iterations of the Perch model we use additional training data (including non-avian taxa; Section 2.1) and new data augmentations and training objectives. We generally find that increasing the difficulty of the classification problem increases the overall quality of the embedding model: To that end, we introduce a novel generalization of mixup (Zhang et al., 2018) that mixes more than two sources (Section 2.1). Additionally, recent BirdCLEF competitions have demonstrated strong results with iterative pseudo-labeling (Kahl et al., 2024). Building on this observation, we introduce a variation of self-distillation (Allen-Zhu & Li, 2022) where a prototype learning classifier (Chen et al., 2019; Heinrich et al., 2025) is used to generate predictions that are used as soft targets for the network's dense linear classifier (Section 2.2 and Section 2.3). Finally, noting that label granularity contributes to transfer learning performance (Section D.1), we add an auxiliary self-supervised loss in the form of source prediction (DIET, Balestriero, 2023), asking the model to predict the source recording of an audio window (Section 2.2 and Section 2.3). A similar objective has been shown to be useful for individual animal identification (Lapp et al., 2025).

Beyond raw performance on bioacoustics benchmarks, the goal of the Perch model is to satisfy properties we believe are important for downstream applications: a relatively small model that can adapt to a wide variety of domains and tasks using linear probing, since this requires fewer computational resources, less machine learning expertise, and less labeled data than full fine-tuning. These constraints enable practitioners to run the model on consumer-grade hardware, enabling robust clustering, nearest-neighbor search[1] or agile modeling (Dumoulin et al., 2025) workflows. To meet these constraints, the Perch 2.0 model is based on EfficientNet-B3 (Tan & Le, 2019), a small model by modern machine learning standards, reducing the processing time of large datasets. We develop an extensive model selection procedure (Section 2.5.1) which evaluates the model in a variety of domain shift and transfer learning tasks using linear probing and retrieval.

## 2 METHODOLOGY

In this section we provide a breakdown of the training data, model architecture, training objectives, and evaluation procedure.

### 2.1 TRAINING DATA

**Data sources** We train on a combination of four labeled audio datasets: Xeno-Canto (Vellinga & Planqué, 2005), iNaturalist (iNaturalist contributors, 2025), the Tierstimmenarchiv (Frommolt, 1996) and FSD50K (Fonseca et al., 2022). The first three contain bioacoustics recordings with species labels while the latter contains a variety of non-bird sounds (Table 1). In total the data contains 14,795 different classes of which 14,597 are species labels and the remaining 198 classes are general sound event classes from FSD50K.

Table 1: Recordings per taxonomic class represented in the training data. *Note that FSD50k contains some coarsely-labeled animal sound classes, which we treat as 'Other' in this table.

|  | Aves | Amphibia | Insecta | Mammalia | Other | Total |
|---|---|---|---|---|---|---|
| Xeno-Canto | 860,701 | 2,260 | 31,971 | 1,323 | 0 | 896,255 |
| iNaturalist | 480,230 | 51,450 | 30,535 | 9,074 | 409 | 571,698 |
| Tierstimmenarchiv | 26,622 | 1,341 | 860 | 4,992 | 44 | 33,859 |
| FSD50k | 0 | 0 | 0 | 0 | 40,966* | 40,966 |
| Total | 1,367,553 | 55,051 | 63,366 | 15,389 | 41,419 | 1,542,778 |

Xeno-Canto and iNaturalist are large citizen-science repositories. The Xeno-Canto data was obtained from the public API[2] and the iNaturalist data was obtained by collecting the 'research-grade'

---

[1] https://search.acousticobservatory.org/
[2] https://xeno-canto.org/explore/api

audio examples reported to the Global Biodiversity Information Facility (GBIF)[3], similarly to Chasmai et al. (2025). Our training datasets for iNaturalist and Xeno-Canto were downloaded in March 2025. The Tierstimmenarchiv (Animal Sound Archive) is a repository from the Berlin Natural History Museum. We collected these recordings from the website[4] in April 2025.

Since Xeno-Canto, iNaturalist and the Tierstimmenarchiv use different taxonomies (species names) we manually mapped the classes from Xeno-Canto and the Tierstimmenarchiv into the iNaturalist classes. We also removed all bat recordings from the datasets (as their vocalizations cannot be represented using the spectrogram parameters we selected).

**Window selection**    All of our data sources contain recordings of variable length, ranging from less than a second to over an hour (with the majority in the 5–150 s range). As a result, the labels for the recordings are 'weak' in the sense that we don't know at which time(s) the species is actually vocalizing if the recording is longer than 5 s. Because our model operates on audio segments of a fixed 5 s size, we must select which windows to feed to the model for training. We explore two window selection methods:

1. Random window selection: Each time a recording is selected for training, produce a uniformly random 5 s audio window. Although this might produce windows in which the target species is not actually heard, the resulting label noise might be of acceptable levels.

2. Energy peak selection: A heuristic that was important in the training of Perch 1.0 where a wavelet transform is used to select windows containing the strongest signal (details in Appendix C). This is based on the assumption that the labeled species is likely the most prominent sound in the recording. We select a 6 s window based on the energy peaks and then select a random 5 s window within this larger window. Similar signal detectors are used in the training of BirdNET (Kahl et al., 2021) and other bioacoustics models (e.g., Sprengel et al., 2016).

**Mixup**    We apply a variation of mixup (Zhang et al., 2018), mixing together different windows of audio in order to create a new composite signal. We generalize the original mixup implementation to more than 2 components as follows: We first choose the number of components by sampling from a beta-binomial distribution, $N \sim \text{BetaBin}(n, \alpha, \beta) + 1$. We then sample weights for each of the components from a symmetric Dirichlet distribution, $\mathbf{w} \sim \text{SymDir}(N, \omega)$. The composite signal is constructed by taking the weighted sum of the components using weights $\mathbf{w} = (w_1, \ldots, w_N)$ and then dividing the signal by $\sqrt{\sum_{i=1}^{N} w_i^2}$ (to ensure that the gain of the mixed audio signal remains unchanged). The parameters $n$, $\alpha$, $\beta$ and $\omega$ are all treated as hyperparameters. Unlike the original mixup implementation, we construct a multi-hot target vector rather than taking a weighted average of one-hot target vectors, as this reflects the fact that all vocalizations in an audio window should be recognized with high confidence irrespective of their loudness.

## 2.2 Model Architecture

The model has three parts: a frontend (which converts raw audio into a spectrogram), an embedding model, and a set of output heads (Figure 2).

**Frontend**    The model frontend consumes monaural audio sampled at 32 kHz and produces a log mel-spectrogram. The frontend takes in audio segments of length 5 s (160,000 samples) and uses a hop and window length of 10 and 20 ms respectively, outputting a total of 500 frames with 128 mel-scaled frequency bins ranging from 60 Hz to 16 kHz. Further details can be found in Section A.2.

**Embedding model**    The embedding model is an EfficientNet-B3 (Tan & Le, 2019), a convolutional residual network with 12 million parameters, utilizing depthwise convolutions to maximize parameter efficiency. Note that this is larger than our original Perch model (which used an EfficientNet-B1 with 7.8 million parameters), reflecting the increased amount of training data. From the embedding model we obtain a *spatial embedding*, $E_S$, of shape $(5, 3, 1536)$ with axes corresponding to time,

---

[3] https://www.gbif.org/

[4] https://suche.tierstimmenarchiv.de/

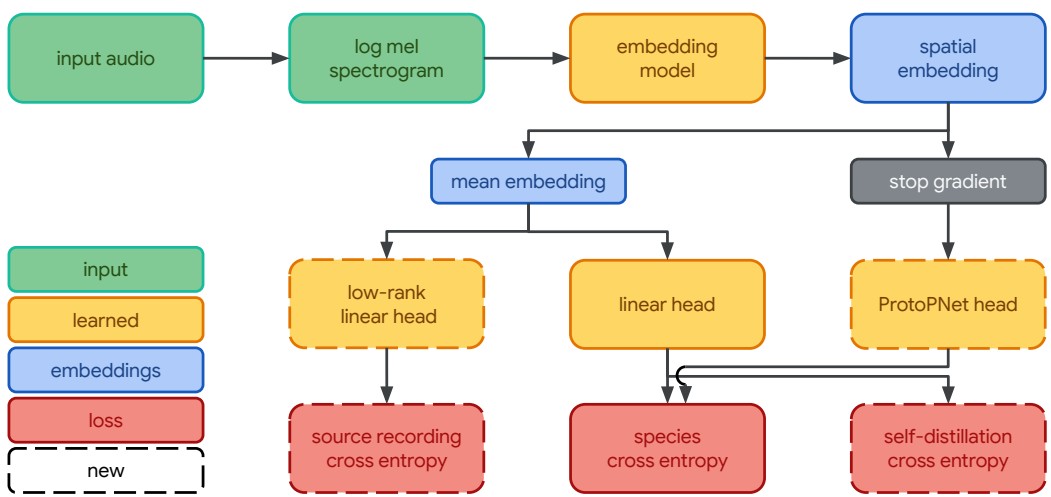

Figure 2: Perch 2.0 model architecture.

frequency and features. We average over the spatial dimensions to obtain a single 1536-dimensional *mean embedding* $E_A$.

**Output heads**    During training, we have three output heads.

1. A *linear classifier* which projects $E_A$ into our 14,795-dimensional space of class labels. Training a linear classifier on our embeddings encourages the final embeddings of our model to be linearly separable for different species which should improve the performance of linear probing.

2. A *prototype learning classifier* which also makes predictions over the class labels. This classifier head was introduced as part of the ProtoPNet model in Chen et al. (2019) and was used in bioacoustics as part of AudioProtoPNet in Heinrich et al. (2025). It takes as input the spatial embeddings, $E_S$. Four prototypes are learned for each class and the final prediction uses the maximum activation across the prototypes.

3. The *source prediction head* is a linear classifier which predicts the (one-hot encoded) source recording from the mean embedding, $E_A$. Our training set has over 1.5 million source recordings so we use a low-rank projection of rank 512. This head is used for the self-supervised source prediction loss from DIET (Balestriero, 2023).

## 2.3 TRAINING OBJECTIVES

Our model is trained by optimizing three separate training objectives.

**Species classification cross entropy**    Following Mahajan et al. (2018) we train the linear classifier using a softmax activation layer and a cross entropy loss using a target vector where each of the $k$ target classes (species) has the value $\frac{1}{k}$. We have found this to train faster than using the more traditional sigmoid binary cross-entropy used in multi-label classification.

**Self-distillation**    The protoype learning classifier is trained for species classification in a similar fashion using a softmax activation function and cross entropy. Following Heinrich et al. (2025); Donnelly et al. (2022) an orthogonality loss is also computed to maximize variation across the prototypes.

A stop-gradient separates the embedding model from the prototype learning classifier so that its gradients do not propagate to the embedding model. Instead, the predictions from the prototype learning classifier are used as soft targets for the linear classifier.

This is a form of self-distillation where the prototype learning classifier is the teacher and the linear classifier is the student (with both sharing the embedding model parameters). Self-distillation is known to improve performance (Allen-Zhu & Li, 2022).

**Source prediction** Source prediction is a simple self-supervised objective: Assign each example in the dataset its own class and train a classifier in supervised fashion as usual using softmax cross entropy. This technique was proposed in Balestriero (2023) in the context of vision and is reliant on data augmentation to force the network to learn salient features. In our case the data augmentation comes in the form of windowing: Longer recordings can produce entirely non-overlapping 5 s windows which the model will need to learn to assign to the same class. That is, the network must predict the source recording given a short 5 s window.

Note that this self-supervised method can just as well be seen as an extremely fine-grained supervised classification problem, which might go some way to explaining its effectiveness.

**Training phases** Our model is optimized in two phases. The linear classifier and source prediction head are trained in both phases. During the first phase we train the prototype learning classifier but we do not use the predictions from the prototype learning classifier as soft targets for self distillation yet. Once the prototype learning classifier is trained we start the second phase of training during which we perform the self-distillation. All losses are minimized using Adam (Kingma & Ba, 2015). For the first and second phases we budget up to 300,000 and 400,000 steps, respectively.

## 2.4 HYPERPARAMETER SELECTION

For hyperparameter selection we apply Vizier (Golovin et al., 2017), a black-box optimization algorithm for machine learning hyperparameter optimization.

For the first phase of training (i.e., without self-distillation) we use Vizier to explore the learning rate, dropout rate applied on the embeddings before the classification heads, source prediction loss weight and mixup hyperparameters. In each phase we run Vizier in two stages, training 100 models in each stage (i.e., 100 models are trained and based on their validation scores Vizier selects a new set of 100 hyperparameters for the second stage). Training for each model took between 20 and 30 hours on a TPUv3-8, depending mainly on the number of mixup signals used.

We select the best model according to the best mean performance on the validation tasks (Section 2.5). We then continue training this model for the second (self-distillation) phase, performing a second sweep with Vizier over the same hyperparameters as before but now adding the self-distillation loss weight as well. We ran both phases of training using random windows and energy peak selection.

During the first phase we found that Vizier preferred models that mixed multiple signals, $N \in \{2, \ldots, 5\}$, had non-negligible source prediction loss weights of in the range $(0.1, 0.9)$[5], and dropout rates in $(0.3, 0.6)$. The better models had $\alpha > \beta$ and a concentration, $\omega$, around $(10, 30)$ for the mixup parameters.

During the self-distillation phase, Vizier preferred models with a small learning rate, little or no mixup (mostly $N = 1$, i.e., no mixing), low source prediction loss, $(0, 0.4)$, and less dropout, $(0, 0.5)$. It assigned high weights to the self-distillation loss, $(1.5, 4.5)$. The hyperparameters of the final models are listed in Section A.1.

## 2.5 EVALUATION

The field of bioacoustics recognized domain shift as an important problem early on (Lasseck & Others, 2013), acknowledging that deployment conditions and tasks might differ from training time. Hence, our model selection (validation) is set up to test several forms of generalization: we consider avian soundscapes (which are qualitatively different from the focal recordings found in our training data); consider tasks other than species identification (e.g., call-type and dialect recognition); and evaluate transfer to species classification of non-avian taxa (bats, marine mammals, mosquitoes, etc.)

---

[5]Note that given the large number of classes in the source prediction task the scale of this loss is higher than that of the species classification loss.

For our model evaluation (testing) we use both BirdSet (Rauch et al., 2025b) and BEANS (Hagiwara et al., 2023). Combined, these benchmarks cover a similar set of domain shifts while still allowing us to make a fair comparison with existing models.

In practice, bioacoustics models like Perch are often used by practitioners who have limited computational resources and machine learning expertise while working on novel problems with little or no labeled data (but potentially large amounts of unlabeled data). In these settings a sensible approach is to embed the data once and then use methods such as clustering, nearest neighbor searches, few-shot learning and agile modeling (Dumoulin et al., 2025). We want to ensure that our model produces embeddings that perform well in these situations, so during both model selection and evaluation we freeze the embedding model.

### 2.5.1 MODEL-SELECTION TASKS

To ensure that our model performs well in the applications described in Section 2.5 we consider three different types of tasks at validation time:

1. We evaluate *pretrained classifier performance* using ROC-AUC on two fully-annotated bird datasets, where the class labels are a subset of the training classes of the Perch model: Powdermill (Denton et al., 2022) (as in Rauch et al. (2025b)) and Caples (Denton et al., 2022). These tasks validate that our model makes useful species predictions out-of-the-box. We make predictions for every 5 s window with a 2.5 s stride and count the prediction as correct when it overlaps with a ground truth annotation.

2. We examine *one-shot retrieval* by selecting random examples from a fully-annotated dataset, ranking the nearest neighbors (using cosine distance) and then computing the ROC-AUC for all examples of that species in the dataset (using all other species as negatives). This is a proxy for our model performance in nearest neighbor searches and clustering applications. We evaluate this task on the BEANS benchmark (Hagiwara et al., 2023)'s detection datasets and the Weldy calltype dataset (Weldy et al., 2024) as well as Powdermill and Caples.

3. Finally, we have *linear transfer tasks* in which we select a random subset of 16 examples per class and compute an embedding for each (by embedding each 5 s window with a stride of 5 s and averaging them). We then train a linear classifier for 10,000 steps on the selected examples using scikit-learn. All remaining examples are used to compute an ROC-AUC score. This follows the procedure applied in Ghani et al. (2023) and evaluates how well our model's embeddings do in few-shot learning and agile modeling setups. We evaluate this task on the BEANS classification tasks, some of the datasets from Ghani et al. (2023) as well as marine datasets such as DCLDE (Palmer et al., 2025), NOAA (NOAA Pacific Islands Fisheries Science Center, 2021) and ReefSet (Williams et al., 2025).

For each task-type, we compute a geometric mean over all relevant datasets to obtain an overall task-performance score. Finally, we compute an overall model quality score by taking the geometric mean of the three task-performance scores. The geometric mean will tend to favor models with less variance in the scores (van Merriënboer et al., 2024).

In summary, the validation tasks for Perch 2.0 were designed to evaluate real world usage of the model on a broad collection of datasets (a total of 19 constituent datasets; refer to Table 2).

### 2.5.2 EVALUATION TASKS

The BirdSet benchmark (Rauch et al., 2025b) consists of six fully-annotated datasets from the continental United States, Hawai'i, Peru and Columbia. BirdSet provides training sets for each of these soundscapes for model fine-tuning. In our evaluations we forgo fine-tuning and simply use the predictions of our prototype learning classifier predictions directly.

The BEnchmark of Animal Sounds (BEANS Hagiwara et al., 2023) provides twelve cross-taxa test tasks, including birds, land and marine mammals, anurans, bats, and insects. Each dataset is divided into train, validation, and test data. We use the train data to train both linear and prototypical probes on the embeddings, but again forgo any fine-tuning of the embedding model. The prototype probing is performed using the same prototype classifier head as described in Section 2.2. In order

Table 2: Summary of Datasets

| | | Train | Validation | | | Test |
| | | | Classify | Retrieval | Transfer | |
| Dataset Name | Citation | | | | | |
| --- | --- | --- | --- | --- | --- | --- |
| Xeno-Canto | Vellinga & Planqué (2005) | ✓ | | | | |
| iNaturalist | iNaturalist contributors (2025) | ✓ | | | | |
| Tierstimmenarchiv | Frommolt (1996) | ✓ | | | | |
| FSD50K | Fonseca et al. (2022) | ✓ | | | | |
| Caples | Denton et al. (2022) | | ✓ | ✓ | | |
| Powdermill | Chronister et al. (2021); Rauch et al. (2025b) | | ✓ | ✓ | | |
| Weldy calltype | Weldy et al. (2024) | | | ✓ | | |
| Ghani transfer[a] | Ghani et al. (2023) | | | | ✓ | |
| DCLDE 2026 | von Benda-Beckmann et al. (2022); Palmer et al. (2025) | | | | ✓ | |
| NOAA PIPAN | Allen et al. (2021); NOAA Pacific Islands Fisheries Science Center (2021); Allen et al. (2024) | | | | ✓ | |
| ReefSet | Williams et al. (2025) | | | | ✓ | |
| BirdSet | Rauch et al. (2025b); Alexander Hopping et al. (2022); Vega-Hidalgo et al. (2023); Kahl et al. (2022a); Clapp et al. (2023); Kahl et al. (2022b) | | | | | ✓ |
| BEANS[b] | Hagiwara et al. (2023) | | | ✓ | ✓ | ✓ |

[a] From this paper only the Godwit Calls, Yellowhammer Dialects and Bats datasets are used. The other datasets overlap with the BEANS benchmark.

[b] We do not use the ESC-50 and Speech Commands datasets for validation.

to accommodate the supersonic vocalizations of bats in BEANS we read the raw audio data as if it was sampled at 32 kHz, effectively pitch shifting the signals to ensure they are in the audible range.

In Section D.2 we compare Perch 2.0 to specialized marine models: SurfPerch (Williams et al., 2025) and Google's Multispecies Whale Model (Harvey et al., 2024).

## 3 RESULTS

We present results comparing Perch 2.0 with a variety of baseline models from the literature on the BirdSet and BEANS benchmarks (described in Section 2.5.2). In Table 3 we present the mean class-mean average precision (cmAP), AUROC, and top-1 accuracy scores on BirdSet across all tasks. For BEANS, we present mean metrics for each of the two task types: mean accuracy on the classification transfer tasks (excluding *esc50* and *speech*), and mean macro-averaged average precision on detection tasks. We include the full breakdown of model performance across benchmark sub-tasks in Appendix D. We also perform ablations on some of the main model design choices in Appendix B.

Although the BirdSet benchmark calculates several scores, we believe that the ROC-AUC score is the most stable and informative (van Merriënboer et al., 2024). On this metric, as well as on the BEANS tasks, Perch 2.0 achieves state-of-the-art performance. Note that this was achieved without any fine-tuning of the embedding model.

Although Perch 2.0's performance with linear probing on BEANS compares favorably to existing models, we note that prototypical probing seems to improve the performance on detection tasks.

Contrary to our experience with Perch 1.0 and other results in the literature, we find that training with random windows performs on par with using energy peak selection. We hypothesize that the self-distillation phase helps address some of the issues with label noise, negating the need for window selection methods.

Table 3: Benchmark results

| | BirdSet | | | | BEANS | | |
|---|---|---|---|---|---|---|---|
| | Method[a] | AUROC | cmAP | Acc | Method[a] | Acc | mAP |
| Audio ProtoPNet-5 | Pre | 0.896 | 0.423 | 0.623 | –[b] | – | – |
| BirdMAE-L[c] | FT | 0.886 | **0.440** | 0.601 | – | – | – |
| BirdMAE-L[c] | PP | 0.886 | 0.409 | 0.521 | – | – | – |
| BirdMAE[d] | AP | 0.865 | – | – | AP | 0.800 | – |
| BEATs NLM[d] | AP | 0.846 | – | – | AP | 0.837 | – |
| AVES-Bio | – | – | – | – | FT | 0.797 | 0.398 |
| BioLingual | – | – | – | – | FT | 0.838 | 0.479 |
| NatureLM-Audio | – | – | – | – | 0 | – | 0.153 |
| Perch 1.0 | Pre | 0.839 | 0.356 | 0.613 | LP | 0.802 | 0.353 |
| Perch 2.0 - Phase I | Pre | 0.902 | 0.431 | 0.642 | LP | 0.835 | 0.426 |
| | | | | | PP | **0.839** | 0.499 |
| Perch 2.0 - Peak-select | Pre | 0.907 | 0.430 | 0.619 | LP | 0.836 | 0.426 |
| | | | | | PP | 0.832 | **0.504** |
| Perch 2.0 - Random | Pre | **0.908** | 0.431 | **0.665** | LP | 0.834 | 0.415 |
| | | | | | PP | 0.836 | 0.502 |

[a] 'Pre' indicates application of a pre-trained classification head, 'FT' indicates model fine-tuning on a train split of the data, 'LP' indicates a linear probe, 'PP' is a prototypical probe, 'AP' is attentive probing, and '0' indicates zero-shot language queries.

[b] For models with a dash (–) in place of a numeric score, one or more scores on a benchmark specific task were not reported.

[c] Note that the BirdSet scores reported for BirdMAE-L are calculated using the scores on the High Sierras Nevada (HSN) subtask, even though this dataset was used by BirdMAE-L for hyperparameter tuning.

[d] As originally reported in Schwinger et al. (2025).

## 4 DISCUSSION

### 4.1 WHY SUPERVISION?

The fields of natural language processing and computer vision have seen a strong trend towards *foundation models* (Bommasani et al., 2021): self-supervised models trained on large amounts of unlabeled data which can be adapted to a variety of downstream tasks with minimal or no fine-tuning. Many other areas of machine learning have tried to emulate this success leading to mixed results (Xu et al., 2024; Cole et al., 2022). As evidenced by the results in this paper, supervision remains dominant in bioacoustics as well. We posit some hypotheses as to why this is the case.

Most successful self-supervised methods depend on using large amounts of unlabeled data and correspondingly large models (Chen et al., 2020a;b, Figure 1). For example, a strong self-supervised model in vision such as DINOv2 (Oquab et al., 2024) was trained on 142 million images. In comparison, Xeno-Canto and iNaturalist are two orders of magnitude smaller. Perhaps bioacoustics will need to unlock significant amounts of diverse, unlabeled data to replicate the success of self-supervised learning in vision.

Self-supervised methods also rely on domain-specific training objectives and data augmentations (Tamkin et al., 2021; Gui et al., 2024) and the performance of models is highly dependent on the chosen data augmentations (Morningstar et al., 2024). In speech, self-supervised models such as wav2vec (Baevski et al., 2020) have shown strong performance but general purpose audio benchmarks such as HEAR (Turian et al., 2022) are still dominated by supervised and semi-supervised models despite several attempts to apply self-supervised methods to general audio problems (Saeed et al., 2021; Al-Tahan & Mohsenzadeh, 2021). Comparisons of bioacoustics models in the literature similarly find that supervised models still outperform self-supervised models (Ghani et al., 2023; Kather et al., 2025). It is possible that work remains on finding the right data augmentations to use in bioacoustics. This domain specificity is reminiscent of similar observations made in the context

of source-free domain adaptation (Boudiaf et al., 2023): there, approaches developed in the context of computer vision problems were not as generalizable to bioacoustics as one would hope.

On the other hand, there are indications that bioacoustics is particularly well-positioned to exploit supervised learning. Results such as those presented in Cole et al. (2022, Figure 2) suggest that if there are enough labeled examples available (hundreds of thousands) then it becomes increasingly difficult to outperform supervised models; we use over 1.5 million labeled recordings in the training of Perch 2.0.

Supervised pre-training benefits particularly from having fine-grained labels (Hong et al., 2024). The performance gap between self-supervised and supervised models becomes bigger as the supervised models have access to finer-grained labels (Cole et al., 2022, Figure 5). Not only does our problem space have over 15,000 classes, but distinguishing between different species within the same genus can be a particularly fine-grained problem. In Section D.1, we demonstrate that reducing the granularity of supervised labels results in worse transfer learning performance for the resulting embeddings.

Furthermore, one could suspect that models trained on primarily avian recordings are likely to fail in transferring to non-avian vocalizations. However, the great diversity of birdsong (Kroodsma, 2004) and the existence of universal mechanisms of sound production in terrestrial vertebrates (Elemans et al., 2015) are both likely to contribute to the successful transfer to a surprisingly large number of bioacoustic domains (Ghani et al., 2023).

## 4.2 CONCLUSION

Though we have previously seen wide transfer of bird vocalization embeddings to a surprising number of bioacoustics tasks, we are confident that the expansion of training targets in Perch 2.0 will make it applicable to an even wider range of problems. We have seen in this work that high quality transfer learning does not require very large models, and that well-tuned supervised models using strong data augmentations and auxiliary training objectives can have strong performance. The simplicity of the final model and the linear separability of its embeddings ensure scalability to large datasets. In particular, transfer and classification on fixed embeddings (as opposed to model fine-tuning) means that embeddings can be computed once and reused for large datasets. This represents a massive savings of resources for e.g. one-shot retrieval through embedding vector search (i.e., 'query-by-example'), which is an important first step to transfer learning through agile modeling (Dumoulin et al., 2025).

## 4.3 FUTURE WORK

The validation procedures used for Perch 2.0 try to close the gap between model evaluation and actual deployment, which is a particularly complex topic (van Merriënboer et al., 2024). We believe more work remains to be done in the development of benchmarks that actually reflect the real world use of bioacoustics models.

Our model's use of source prediction also opens up a clear avenue for semi-supervised learning which can be used to fill gaps in labeled training data (e.g., for taxa that have relatively few labeled examples such as mammals).

The success of supervised learning also raises the question as to whether we can construct other classification tasks using the metadata that is available for many recordings (e.g., the time of day, season, location).

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

## A  MODEL DETAILS

### A.1  HYPERPARAMETERS

For the first phase of training, the best performing model had the following hyperparameters:

- Dropout rate: 0.49
- Source prediction loss weight: 0.11
- Mixup parameters: $\alpha = 91.3, \beta = 100, \omega = 1, n = 2$
- Learning rate: $6.41 \cdot 10^{-4}$

For the second phase of training, the selected hyperparameters were:

- Self-distillation loss weight: 4.22
- Dropout rate: 0
- Source prediction loss weight: 0
- Mixup parameters: $n = 0$ (no mixup)
- Learning rate: $3.20 \cdot 10^{-6}$

## A.2 FRONTEND

Our frontend outputs mel-scaled log-spectrograms, taking in 5 s of audio at 32 kHz (160,000 samples). It uses a hop length of 10 ms (320 samples) and a window length of 20 ms (640 samples). The FFT window is set to 1,024 samples for computational efficiency. The frames are uncentered (i.e., the first frame begins at the first sample) and a Hann window is used. We calculate the energy (magnitude) spectrogram (so not a power spectrogram). The mel-scale is calculated using the HTK formula. Similar to SciPy's STFT implementation the output is scaled by the reciprocal of the sum of the window values. After the calculation of the mel-spectrogram we apply a logarithm with a floor of $10^{-5}$ and then multiply the output by 0.1.

## B ABLATIONS

To demonstrate the relative performance of training choices and data availability, we provide a collection of ablations (Table 4). For the ablations, we modify the Perch 2.0 – Phase I model, with random window selection, to demonstrate the relative role of data, mix-up, EfficientNet architecture size, and index prediction. We provide the BirdSet ROC-AUC and mean of all BEANS (LP) task scores for each model, as well as the geometric mean of these BirdSet and BEANS scores.

The models are all point-wise modifications of the Perch 2.0 Phase I configuration, ablating training data (Xeno-Canto only), mixup (2, 4), architecture (B1, B5), and removing index prediction.

We observe somewhat higher overall test performance from the B5 architecture (indicating headroom for additional model capacity), but otherwise all architectural modifications lead to somewhat worse performance. Note that mixup=4 provides a nice boost on BirdSet, but commensurate regression on BEANS tasks, perhaps because many of these tasks are single-label and the improved performance on mixed audio signals is irrelevant.

Table 4: Ablation Results.

|  | BirdSet | BEANS (LP) | Geometric mean |
|---|---|---|---|
| Perch 2.0 - Phase I | 0.902 | 0.664 | 0.774 |
| *Xeno-Canto only* | -0.024 | -0.008 | -0.015 |
| *2 mixup components* | -0.015 | +0.003 | -0.005 |
| *4 mixup components* | +0.002 | -0.004 | -0.002 |
| *EfficientNet B1* | -0.003 | -0.015 | -0.010 |
| *EfficientNet B5* | -0.003 | +0.013 | +0.006 |
| *No index prediction* | -0.008 | -0.003 | -0.005 |

## C PEAK-FINDING

A mel-spectrogram of the recording is constructed using a window size of 80 ms and a hop of 10 ms. The magnitudes are log-scaled (using a floor of 0.01) and then scaled by 0.1.

Then, a two-step denoising process is applied: For each frequency bin the mean and standard deviation of the log-magnitudes is calculated across time. Any values which are greater than the mean plus 1.5 standard deviations are discarded. A second mean and standard deviation is calculated using the remaining values. This second mean and standard deviation are used to select signal, which are all values that lie above the mean plus 0.75 standard deviations. The signal is shifted by this second mean, and the rest is discarded.

Finally, the magnitudes of the denoised spectrogram are summed across the frequency bins. SciPy's `signal.find_peaks_cwt` is then used to find peaks ranging between 0.5 and 2 s using 10 wavelet filters. We calculate the total value of the summed magnitudes in the 600 ms window surrounding the peak. If this total value is less than 1.5 times the mean frequency-summed magnitude over the entire recording the peak is discarded. The remaining peaks are sorted by their summed magnitudes and only the top 5 are kept.

If a recording is less than 6 s, it is padded with zeros before applying the peak finding. If not a single peak is found, we simply select the first 6 s of the recording.

# D ADDITIONAL RESULTS

Table 5: BirdSet per-dataset results.

| | PER | NES | UHH | HSN | NBP | SSW | SNE | *Mean* |
|---|---|---|---|---|---|---|---|---|
| | | | | ROC-AUC | | | | |
| Audio ProtoPNet-5 | 0.790 | 0.930 | 0.870 | **0.920** | 0.930 | 0.970 | 0.860 | *0.896* |
| BirdMAE-L (FT) | **0.820** | 0.910 | 0.820 | 0.900 | 0.940 | 0.930 | 0.880 | *0.886* |
| BirdMAE-L (PP) | **0.820** | 0.930 | 0.830 | 0.900 | 0.920 | 0.940 | 0.860 | *0.886* |
| BirdMAE (AP) | 0.782 | 0.886 | 0.815 | 0.891 | 0.922 | 0.924 | 0.838 | *0.865* |
| BEATs NLM (AP) | 0.730 | 0.892 | 0.807 | 0.845 | 0.901 | 0.932 | 0.811 | *0.846* |
| Perch 1.0 | 0.700 | 0.900 | 0.760 | 0.860 | 0.910 | 0.910 | 0.830 | *0.839* |
| Perch 2.0 - Phase I | 0.783 | 0.948 | 0.896 | 0.896 | **0.941** | 0.968 | 0.885 | *0.902* |
| Perch 2.0 - Peak-Select | 0.802 | 0.948 | 0.901 | 0.896 | **0.941** | **0.974** | **0.887** | *0.907* |
| Perch 2.0 - Random | 0.786 | **0.953** | **0.912** | 0.915 | 0.933 | 0.973 | 0.883 | ***0.908*** |
| | | | | cmAP | | | | |
| Audio ProtoPNet-5 | 0.300 | 0.380 | 0.310 | 0.540 | 0.680 | 0.420 | 0.330 | *0.424* |
| BirdMAE-L (FT) | **0.350** | **0.410** | 0.300 | **0.550** | **0.720** | 0.410 | 0.340 | ***0.440*** |
| BirdMAE-L (PP) | 0.310 | 0.380 | 0.300 | 0.490 | 0.690 | 0.380 | 0.300 | *0.407* |
| Perch 1.0 | 0.180 | 0.390 | 0.270 | 0.450 | 0.630 | 0.280 | 0.290 | *0.356* |
| Perch 2.0 - Phase I | 0.230 | 0.396 | 0.368 | 0.522 | 0.682 | 0.464 | **0.351** | *0.431* |
| Perch 2.0 - Peak-Select | 0.255 | 0.382 | 0.375 | 0.504 | 0.695 | 0.455 | 0.342 | *0.430* |
| Perch 2.0 - Random | 0.232 | 0.403 | **0.380** | 0.533 | 0.661 | **0.469** | 0.341 | *0.431* |
| | | | | Top-1 Accuracy | | | | |
| Audio ProtoPNet-5 | 0.590 | 0.520 | 0.490 | 0.650 | 0.710 | 0.660 | 0.740 | *0.623* |
| BirdMAE-L (FT) | 0.600 | 0.520 | 0.420 | 0.640 | 0.720 | 0.700 | 0.610 | *0.601* |
| BirdMAE-L (PP) | 0.590 | 0.470 | 0.360 | 0.380 | 0.690 | 0.620 | 0.540 | *0.521* |
| Nature LM-Audio | **0.620** | 0.470 | **0.600** | 0.580 | 0.660 | – | 0.760 | – |
| Perch 1.0 | 0.480 | **0.660** | 0.570 | 0.580 | 0.690 | 0.620 | 0.690 | *0.613* |
| Perch 2.0 - Phase I | 0.534 | 0.537 | 0.539 | 0.629 | 0.703 | 0.778 | 0.776 | *0.642* |
| Perch 2.0 - Peak-Select | 0.528 | 0.504 | 0.462 | 0.620 | 0.677 | 0.782 | 0.763 | *0.619* |
| Perch 2.0 - Random | 0.535 | 0.562 | 0.595 | **0.659** | **0.724** | **0.789** | **0.791** | *0.665* |

Table 6: BEANS per-dataset results on tasks measuring accuracy. Mean excludes esc50 and speech.

| | esc50 | watkins | bats | cbi | dogs | humbugdb | speech | Mean |
|---|---|---|---|---|---|---|---|---|
| AVES-Bio | 0.773 | 0.879 | 0.748 | 0.598 | 0.950 | 0.810 | **0.964** | 0.797 |
| BioLingual (FT) | **0.908** | 0.894 | 0.766 | 0.744 | **0.971** | 0.817 | – | **0.838** |
| NatureLM-Audio | 0.820 | 0.788 | – | 0.778 | – | 0.114 | – | – |
| BirdMAE (AP) | – | 0.889 | 0.752 | 0.662 | 0.889 | 0.809 | – | 0.800 |
| BEATs NLM (AP) | – | **0.904** | 0.748 | 0.769 | 0.942 | **0.820** | – | 0.837 |
| Perch 1.0 | 0.800 | 0.855 | 0.718 | 0.757 | 0.942 | 0.739 | 0.853 | 0.802 |
| Perch 2.0 - Phase I | 0.892 | 0.900 | 0.764 | 0.791 | 0.964 | 0.756 | 0.776 | 0.835 |
| Perch 2.0 (Peak., LP) | **0.908** | 0.900 | 0.774 | 0.789 | 0.957 | 0.758 | 0.789 | 0.836 |
| Perch 2.0 (Peak., PP) | 0.890 | 0.858 | **0.815** | 0.785 | 0.935 | 0.768 | 0.838 | 0.832 |
| Perch 2.0 (Rand., LP) | 0.895 | 0.885 | 0.766 | 0.792 | 0.964 | 0.762 | 0.801 | 0.834 |
| Perch 2.0 (Rand., PP) | 0.875 | 0.870 | 0.804 | **0.793** | 0.942 | 0.770 | 0.827 | 0.836 |

## D.1 LABEL GRANULARITY

We perform an additional experiment to demonstrate the importance of label granularity for transfer learning tasks.

Table 7: BEANS per-dataset results on tasks measuring mAP. 'Peak' and 'Rand' indicate peak-selected vs. random window distilled models, 'LP' indicates linear probe, and 'PP' indicates prototypical probe.

|  | dcase | enabirds | hiceas | rfcx | hainan gibbons | Mean |
|---|---|---|---|---|---|---|
| AVES-Bio | 0.392 | 0.555 | 0.629 | 0.13 | 0.284 | 0.398 |
| BioLingual (FT) | **0.475** | 0.688 | **0.677** | 0.178 | 0.376 | 0.479 |
| NatureLM-Audio | 0.058 | 0.314 | 0.336 | 0.025 | 0.005 | 0.148 |
| Perch 1.0 | 0.283 | 0.603 | 0.502 | **0.232** | 0.146 | 0.353 |
| Perch 2.0 - Phase I | 0.370 | 0.656 | 0.516 | 0.153 | 0.434 | 0.426 |
| Perch 2.0 (Peak., LP) | 0.373 | 0.671 | 0.497 | 0.137 | 0.452 | 0.426 |
| Perch 2.0 (Peak., PP) | 0.457 | **0.764** | 0.585 | 0.200 | **0.516** | **0.504** |
| Perch 2.0 (Rand., LP) | 0.377 | 0.661 | 0.492 | 0.141 | 0.406 | 0.415 |
| Perch 2.0 (Rand., PP) | 0.469 | 0.751 | 0.575 | 0.200 | 0.515 | 0.502 |

For this experiment, we train a model on Xeno-Canto only. No prototype linear classifier is trained; architecturally this is the same as Perch 1.0, but trained on increased Xeno-Canto data relative to the earlier model. (Note that the total number of classes for this new model is slightly smaller than Perch 1.0; this is due to the removal of some species from Xeno-Canto due to poaching concerns.)

We observe steady degradation of transfer learning performance as the labels are made more granular.

We additionally observe that the new model trained on only Xeno-Canto outperforms Perch 1.0, demonstrating the impact of accumulating new supervisory data over time.

Table 8: Label granularity

| Model | Num Classes | BEANS Acc. | BEANS mAP |
|---|---|---|---|
| Perch 1.0 | 10932 | 0.809 | 0.353 |
| XC: Species | 10906 | **0.837** | **0.398** |
| XC: Genus | 2398 | 0.820 | 0.389 |
| XC: Family | 249 | 0.761 | 0.318 |
| XC: Order | 41 | 0.608 | 0.217 |

Transfer learning performance of models trained on Xeno-Canto alone, with training labels converted to different levels of granularity.

## D.2 Few shot learning for underwater audio results

In marine bioacoustics, association of sounds with species is much more challenging given the general lack of visual confirmation. In this context, few-shot learning and transfer learning are valuable tools that allow for quick iteration as new sounds are discovered.

**Datasets** To establish Perch 2.0's performance for underwater audio tasks, we evaluate its performance on three marine audio validation sets: NOAA PIPAN, ReefSet, and DCLDE 2026. The DCLDE 2026 dataset is further broken down into three label sets for evaluations:

- Species: humpback, orca, abiotic, and undetermined biological sounds. The specific ecotype annotations for orcas are combined into a single label for this task.
- Ecotype: orca-NRKW, -OKW, -SAR, -SRKW, and -TKW. Ecotypes are locally adapted populations with distinct traits, e.g., NRKW refers to northern resident killer whales (orcas) which live in the northeast part of the Pacific Ocean.
- Known species: humpback and orca.

The NOAA PIPAN audio data comes from the NOAA Passive Acoustic Archive (NOAA Pacific Islands Fisheries Science Center, 2021). The labeled segments were extracted from annotations

provided in Allen et al. (2021; 2024) in addition to annotations provided by NOAA. The label classes in the NOAA PIPAN evaluation set are at the species level for the following baleen species: common minke whale, humpback whale, sei whale, blue whale, fin whale, and Bryde's whale. Additional label classes include anthropomorphic noise, unknown whale species, and 'other'. The ReefSet data are described in detail in Williams et al. (2025), but include a mix of biological reef noises (such as croaks, crackles, growls), classes for some species/genera (e.g., damselfish, dolphins, and groupers), as well as anthropomorphic noise classes and waves.

**Method**    We compare the performance of the mean embeddings generated by Perch 2.0 versus Perch 1.0, SurfPerch (Williams et al., 2025), and the Google Multispecies Whale model (Harvey et al., 2024; Allen et al., 2024) on few-shot classification using ROC-AUC computed from $k = 16$ examples per class in Table 9. We use the same transfer learning protocol with linear probing as described in Section 2.5.1, i.e., we evaluate the embeddings of these models in a few-shot classification setup and do not use the predictions of the models directly.

Note that the published SurfPerch model was trained on the ReefSet data, and similarly a large portion of the NOAA PIPAN labeled audio data was used to train the Multispecies Whale model (but the Multispecies Whale model does not include classes or label sets for sei whales, anthropomorphic noise, or unknown whale). The data for DCLDE 2026 is previously unseen for all models, although the Multispecies Whale model was trained on killer whales. The scores in Table 9 might differ slightly from previously reported results in the literature due to differences in the linear probe estimation implementation[6]. Perch 2.0's training data includes a few dozen cetacean recordings, but these were mostly phone recordings made above water and not reflective of underwater hydrophone recordings.

Table 9: Marine learning transfer tasks

| Model | DCLDE 2026 | | | | | | NOAA PIPAN | | Reefset | |
| | Species | | Ecotype | | Known Bio Species | | | | | |
| | LP | CLS | LP | CLS | LP | CLS | LP | CLS | LP | CLS |
|---|---|---|---|---|---|---|---|---|---|---|
| Multispecies Whale | 0.914 | – | 0.821 | – | 0.954 | 0.612 | 0.917* | – | 0.855 | – |
| SurfPerch | 0.947 | – | 0.903 | – | 0.984 | – | 0.899 | – | **0.986*** | – |
| Perch 1.0 | 0.968 | – | 0.931 | – | 0.981 | – | 0.905 | – | 0.970 | – |
| Perch 2.0 | **0.977** | – | **0.945** | – | **0.989** | – | **0.924** | – | 0.981 | – |

ROC-AUC of few shot transfer learning of Perch 2.0 against Perch 1.0 and pretrained models for Multispecies Whale and SurfPerch for $k = 16$ samples per class for training. (*) indicates the dataset was in the training data for the model.

**Results**    We find that Perch 2.0 outperforms the published comparison models on the linear transfer tasks for cetacean species (DCLDE 2026 species, DCLDE 2026 ecotypes, DCLDE 2026 species-known bio and NOAA PIPAN whales). On ReefSet, Perch 2.0 has an ROC-AUC of 0.981 versus 0.986 from SurfPerch (which was trained directly on ReefSet). These results show that the Perch 2.0 has outstanding performance for transfer learning even in the marine acoustic environments, and thus show that when doing transfer learning or agile modeling, using Perch 2.0 as the embedding model would be expected to provide a more efficient and high quality search and classification workflow for similar types of sounds. For the DCLDE data, if we use logits from directly applying the Google Multispecies model, the AUC-ROC is 0.612, but if using the embeddings from this model for few-shot learning, the performance jumps to 0.954. The discrepancy in performance in those two approaches demonstrates the value of pretrained models as foundational models that can be efficiently fine-tuned for specific domains and classes, even if the model's supported label classes for logits have lower performance. With that said, Perch 2.0 demonstrated superior performance against the whale-specific model in the linear probe, and thus would be recommended for generating embeddings for cetacean-classification tasks.

---

[6]In particular, Williams et al. (2025) evaluated the SurfPerch model on ReefSet by training a linear layer using mini-batches and the Adam optimizer. In our experiments we use scikit-learn's `LogisticRegression` class, which uses the L-BFGS optimizer and weight decay by default.

