# OpenReview forum: "Perch 2.0: The Bittern Lesson for Bioacoustics"
_ICLR.cc/2026/Conference — ICLR 2026 Conference Desk Rejected Submission_

### Official Review · Reviewer_kWia · 2025-10-29

**Soundness:** 1
**Presentation:** 3
**Contribution:** 2
**Rating:** 4
**Confidence:** 4

**Summary:**

This paper introduces Perch 2.0, an update to a pre-trained supervised model for (bird) bioacoustics. The key contributions include expanding the training data from exclusively bird species to a large multi-taxa dataset and adding a combination of existing training techniques: a prototype-based self-distillation, and an auxiliary source-prediction task. Perch 2.0 achieves state-of-the-art performance on different task types (retrieval, linear probing and pretrained out-of-the-box peformance) on the BirdSet and BEANS benchmarks. The paper also provides hypotheses for why supervised learning remains a dominant paradigm in the bioacoustics domain.

**Strengths:**

1. *Clarity and motivation*: The paper is well-written and very easy to follow. The work is well-motivated by addressing real-world challenges faced by practitioners, such as the need for strong, generalizable embeddings from smaller models that do not require extensive fine-tuning.
2. *Methodological combination:* The work combines existing techniques (self-distillation, source prediction, prototype learning) into a single training framework. This combination is well-suited to the problem of fine-grained, multi-class bioacoustic classification.
3. *Comprehensive and practical evaluation:*  By testing the model on a diverse set of tasks, including pretrained out-of-the-box performance, one-shot retrieval, and few-shot linear, the authors mirror the practical ways such models could be deployed, providing a robust evaluation (and for further use in the future).

**Weaknesses:**

While the proposed method combination is interesting and the results are strong, the paper is limited by a lack of empirical validation. The core issue is an absence of ablation studies, which makes it impossible to attribute the performance gains to the specific contributions claimed by the authors (method-based, data-based, etc.).

**1. Confounded contributions and lack of ablations:**

The core weakness is that the paper simultaneously introduces multiple changes to the previous model (a larger model backbone, more training data from new taxa, and several new training objectives,…). The central question of why the model has improved remains unanswered. For the work to be a strong scientific contribution, it would need to include ablations that isolate the impact of each component. Examples include:

- *Data vs. method:* What is the performance gain from the new multi-taxa data alone, when training with the original Perch 1.0 methodology?
- *Methodological components:* What is the individual contribution methods? The paper introduces a novel combination of methods but provides no empirical evidence that they are the actual drivers of the performance increase (e.g., mixup)
- *Model scale:* What is the impact of moving from an EfficientNet-B1 to a B3 backbone? This choice feels arbitrary without a justification or comparison.

**2. Insufficient baselines and reproducibility issues:**

- *Missing baselines*:The evaluation would benefit from a category of baselines: general-purpose audio models pretrained on large datasets like AudioSet. Comparing would help contextualizing the performance of a domain model and support claims of SOTA performance.
- *No code or release plan*: For a paper centered on a new model and its performance on new data compositions, the complete absence of code, model weights, or a concrete plan for their release degrades the quality. The paper does not state when or how the community will be able to access these resources, which limits the work's reproducibility (at least in the context of the review process now) and potential impact.

**Questions:**

1. The paper's main weakness is the confounding of variables (new data, new architecture, new training objectives). Could you provide ablation studies that disentangle these factors? Specifically, can you demonstrate the individual performance contributions of the expanded dataset versus the new training objectives (self-distillation, source prediction, etc.) to understand what is truly driving the reported improvements?
2. Could you justify the architectural choice of EfficientNet-B3 over other backbones?
3. Could you please provide a concrete plan, including a timeline, platform and format, for the public release of the training/evaluation code, the model weights, and the exact data splits used in your experiments?

---

> ### Author Response · Authors · 2025-11-26
>
> We thank Reviewer kWia for their feedback.
>
> > What is the performance contribution of the additional data?
>
> If we limit our phase I training to only Xeno-Canto, we find that the performance on BirdSet drops to 0.878 (compared to 0.902), and a score of 0.656 on BEANS (compared to 0.698). This shows that the inclusion of these datasets is in fact one of the strongest drivers of performance.
>
> > What is the performance contribution of generalized mixup?
>
> We re-trained our Phase I model with k = 2 and achieved a score of 0.887 on BirdSet (compared to 0.902), and a score of 0.667 on BEANS (compared to 0.698), which shows that increasing k beyond 2 does improve performance.
>
> > What is the performance contribution of the bigger model scale?
>
> The ablations we have added in the appendix show that moving to B3 from B1 is the second-largest contributor to the increased performance of the design choices we ablated. This is not surprising: as the amount of training data has increased compared to Perch 1.0, we are able to utilize more compute capacity.
>
> Our ablations show that increasing capacity even further (to a B5 model) would have further increased the test scores (with BEANS benefiting in particular). However, it is not surprising that our model selection procedure did not select the larger B5 model. Our model selection tasks evaluate one-shot retrieval and linear transfer (with 16 examples) with the aim of selecting models that are able to adapt with no or very little training data. We believe that this favours models that can be used in real downstream applications (e.g., agile modelling or vector search). These tasks weren’t necessarily designed to select models that perform well on BEANS (where most tasks have thousands of training examples available, rather than 16, which would allow it to benefit from larger embedding/models).
>
> > What is the performance contribution of self-distillation?
>
> The impact of self-distillation can be seen in Table 3, which compares the phase I model (i.e., before self-distillation) with two different ways of self-distilling (namely, using either random windows or using the windows selected using energy-peaks).
> The increase in performance is arguably quite minor: We see a small jump in BirdSet performance (from 0.902 to 0.908/0.907 in ROC-AUC) and a small jump in BEANS mAP (e.g., from 0.499 to 0.504/0.502 for prototypical probing).
>
> > What is the performance contribution of source prediction?
>
> We re-trained our Phase I model without source prediction and achieved a score of 0.894 on BirdSet (compared to 0.902), and a score of 0.661 on BEANS (compared to 0.698).
>
> > Plan for code and model weights release.
>
> An open-weights Perch 2.0 model checkpoint is already distributed on Kaggle. We cannot provide the link at the moment to avoid breaking anonymity, but we commit to do so in the camera-ready revision. Our training code is intertwined with a variety of libraries that have not been publicly released, which makes it difficult for us to provide a fully-functioning training pipeline, but we commit to releasing reference implementations for the important parts of our model jointly with the camera-ready revision (e.g., generalized mixup implementation, protopnet head, and so on).
>
> > Question: why EfficientNet-B3 over other backbones?
>
> A full exploration of all possible architectures was outside of the scope of this work, but we can highlight some of the reasons why we left our choice of EfficientNet backbone unchanged. Firstly, EfficientNet models have been very successful in bioacoustics as evidenced by their repeated use in BirdCLEF competitions. Moreover, many transformer-based models require more complicated readout layers which makes it harder to deploy them in downstream applications that benefit from linearly separable embeddings (e.g., for clustering) and agile modeling (which uses linear classifiers).
> We believe it could be interesting to explore the ConvNeXt V2 architectures--as was done in Rauch et al.’s (2025) BirdSet paper and Heinrich et al.’s (2024) AudioProtoPNet paper--but we doubt that changing to this backbone will have a significant payoff.

---

### Official Review · Reviewer_fms8 · 2025-10-30

**Soundness:** 1
**Presentation:** 2
**Contribution:** 1
**Rating:** 2
**Confidence:** 4

**Summary:**

This paper presents Perch 2.0, a supervised audio classification model that incorporates three contributions: (1) a novel mixup augmentation variant, (2) a self-distillation mechanism using prototype-based learning, and (3) an auxiliary self-supervised objective. The model adopts an EfficientNet backbone with modifications to the training pipeline and is trained on a combined dataset comprising four existing datasets. The authors report state-of-the-art performance on benchmark evaluations, demonstrating improvements over previous approaches.

**Strengths:**

- **Clear presentation**: The paper is well-written and easy to follow.
- **Comprehensive evaluation framework**: The inclusion of different model selection tasks in the evaluation is nice, as it helps to identify both strengths and limitations of the model.
- **Pragmatic focus on supervised learning**: The decision to focus on supervised learning rather than following the current trend toward self-supervised methods is commendable. This work demonstrates that supervised approaches remain competitive and that self-supervised methods have not yet clearly surpassed them. This is an important contribution that encourages balanced research and prevents the field from prematurely abandoning promising supervised techniques.

**Weaknesses:**

**Unclear novelty and insufficient differentiation from prior work**

The authors claim several contributions, including a novel mixup procedure, a self-distillation process, and a self-supervised auxiliary loss. However, the paper lacks clarity in distinguishing what constitutes genuinely novel contributions versus adaptations of existing techniques. For example, while the authors propose generalizing mixup to more than two components, they do not adequately discuss related work that already explores multi-component mixing strategies (e.g.,  [1]), nor do they clearly articulate how their approach differs from or improves upon these existing methods.

This issue extends to other claimed contributions as well. The paper reads more as a collection of independently developed techniques combined together rather than a cohesive framework with a clear design rationale.

For each component, it should be explicitly stated:

- What is the actual novel contribution beyond existing literature?
- What is the conceptual or empirical rationale for including this specific component?
- How does it differ from and improve upon related work?

Without this clarity, it becomes difficult to assess the true technical contributions of the paper and understand the principled reasoning behind the proposed design choices.

[1] Jang, Junwoo, Jungwoo Han, and Jinwhan Kim. "K-mixup: Data augmentation for offline reinforcement learning using mixup in a Koopman invariant subspace." *Expert Systems with Applications* 225 (2023): 120136.

**Missing rationale for key design choices**

Beyond the novelty concerns, the paper lacks sufficient justification for its key design choices. While the authors incorporate multiple techniques from related work, they do not provide clear reasoning for *why* these specific design decisions should be beneficial or how they contribute to improved performance.

Critical questions remain unanswered throughout the paper:

- What is the rationale behind the specific mixup design? Why should this particular formulation be advantageous over alternatives?
- What motivates the particular construction of the output heads? What benefits does this architecture provide?
- More broadly, what is the underlying principle or insight that guides these design decisions?

This lack of motivation makes it difficult to understand whether the design choices are principled or simply empirically driven. The issue is compounded by an imbalanced presentation: while crucial design decisions lack proper justification, the paper dedicates excessive detail to minor implementation choices that appear neither novel nor particularly important. The authors should prioritize explaining the conceptual motivation behind their key contributions and reduce emphasis on less significant details.

**Absence of empirical evidence and ablation studies**

While the paper introduces numerous design choices, the evaluation is extremely limited and lacks any ablation studies to investigate these decisions. There is not a single experiment that systematically examines the individual contributions of the proposed components, making it impossible to assess which design choices actually benefit model performance and to what extent.

Given that this paper presents a combination of several techniques, this represents my strongest concern. When theoretical justification for a method is absent, rigorous empirical validation through ablation studies becomes essential. For example, there should be experiments that compare the generalized mixup variant against the vanilla mixup baseline to demonstrate the specific improvements their formulation provides.

This requirement applies to every key design choice in the paper. Without such ablations, the paper essentially presents an uncontrolled combination of techniques with no evidence about what actually drives the reported improvements. The claimed contributions cannot be properly validated, and readers have no guidance on which components are essential versus which might be redundant or even detrimental in other contexts.

**Minor Issues:**

- **Inconsistent caption placement**: The caption for Figure 1 is placed below the figure, while the caption for Figure 2 appears above. Caption placement should be consistent throughout the paper, preferably below all figures, following standard conventions.
- **Improper citations**: Several references cite arXiv preprints when the work has actually been published at peer-reviewed conferences. Authors should update citations to reflect the published versions where available.
- **Poor figure integration**: Multiple figures are included in the paper, but are never referenced in the main text. All figures should be explicitly referenced and discussed. Additionally, the purpose of Figure 1 is unclear. It appears to simply show an image of a bird without providing meaningful context or contributing to understanding the method. If figures do not serve a clear purpose, they should be removed or replaced with more informative visualizations.

**Questions:**

See Weaknesses.

---

> ### Author Response · Authors · 2025-11-26
>
> We thank Reviewer fms8 for their feedback.
>
> > Unclear novelty, insufficient differentiation from prior work
>
> We want to clarify that our paper doesn’t claim to make any major architectural or algorithmic innovations. The novel aspects of our work (generalized mixup, source prediction) are relatively straightforward extensions from previous work (MixUp, DIET) to the setting of audio/bioacoustics.
>
> Instead, the contribution of our work is in the careful collection and cleaning of data, thorough exploration of the space of data augmentations, model architectures, and training objectives. The result is a “proof of existence” for a simple, small CNN model that is trained in a relatively straightforward supervised manner, followed by a careful analysis and literature review of the reasons why our approach was successful. The final model outperforms far larger and more complex models, which provides a valuable data point to the bioacoustics community as to where future research and resources should be allocated.
>
> Our findings go against the general trend in bioacoustics (and machine learning more broadly) to develop complex self-supervised models, and as such we believe our findings are a valuable addition to this conversation.
>
> > Missing rationale for key design choices
>
> We hope that the ablation studies we have included in the appendix empirically justify the design choices that we made.
>
> > Ablation studies, including a comparison of generalized mixup and vanilla mixup
>
> We note that the generalized mixup is in fact a relatively straightforward extension of the usual mixup as the Dirichlet distribution is the natural generalization of the beta distribution normally used for MixUp; thus if we set k = 2 then we retrieve the original mixup formulation.
>
> We re-trained our Phase I model with k = 2 and achieved a score of 0.887 on BirdSet (compared to 0.902), and a score of 0.667 on BEANS (compared to 0.698), which shows that increasing k beyond 2 does improve performance.
>
> > Inconsistent caption placement, improper citations
>
> Thank you for flagging this. We moved the caption in Figure 2 and made another pass through the paper and updated the bibliography where needed.
>
> > Figures are in the paper that are not referenced in the main text.
>
> As Figure 1 relates to the paper title and its caption is entirely self-contained, we are unsure as to where it should be referenced in the text. We added a reference to Figure 2.
>
> > The purpose of Figure 1 is unclear.
>
> The purpose of figure 1 is to explain the wordplay in the paper’s title: “the bittern lesson”. As the caption states, “The Bitter Lesson” is an essay from Sutton which argued that “general methods that leverage computation are ultimately the most effective” and that is essentially futile for researchers to “seek to leverage their human knowledge of the domain”.
>
> We see some parallels between Sutton’s insights and our own findings: Perch 2.0 is a simple model that used more computation (more data, more careful hyperparameter tuning) and as a result outperformed far more complex audio models.
>
> As a nod to Sutton’s essay and its applicability to the domain of bioacoustics, we added a photo of a bittern. Although we agree that the figure isn’t strictly necessary and could be removed, we would like to argue in favour of keeping it (because birds are cool).

---

### Official Review · Reviewer_TSZg · 2025-10-30

**Soundness:** 3
**Presentation:** 3
**Contribution:** 1
**Rating:** 4
**Confidence:** 4

**Summary:**

The paper introduces Perch 2.0, an updated version of the bioacoustic
classification model Perch. Perch 2.0 includes more diverse training data,
slightly modified architecture, and auxiliary loss functions. The training data
includes broader species coverage from Xeno-Canto, iNaturalist, Tierstimmenarchiv,
and FSD50k. The model architecture is now based on EfficientNet-B3 (compared with
EfficientNet-B0 in Perch 1.0) and includes a prototype-learning classifier head,
a source-prediction head, and a linear classification head. Training is conducted
in two stages: First, only the linear-classification and source-prediction heads
are used to compute the loss; then, the prototype-learning head is used to provide
soft targets for the linear-classification head, a form of self-distillation.
Hyperparameter tuning is conducted with Vizier using multiple
validation tasks. The selected model is then evaluated on BirdSet and BEANS,
achieving state-of-the-art results on both datasets. Subsequently, it is claimed
that "supervision remains dominant in bioacoustics" as shown by the strong
performance of Perch 2.0 and the benefit of fine-grained labels.

**Strengths:**

- The paper is well-written and easy to follow.
- The model achieves state-of-the-art results on multiple datasets.

**Weaknesses:**

While the developed model shows strong performance, the question remains: what
contributes to its strong performance? As multiple changes were made compared with
the BirdSet and Perch 1.0 baselines, it is hard to assess the importance of each
individual change. Most importantly, it is unclear how much the additional
training data contributes to the performance increase relative to the
architectural changes and auxiliary losses. Ablation studies could help clarify
this. This is especially important because the authors conclude that
"supervision remains dominant in bioacoustics"; however, it is unclear whether this
conclusion still holds if Perch 2.0 is trained on the same data as BirdMAE.

In addition, the following weaknesses were identified:
- Reproducibility is hindered because no training code is provided, and no fixed sets
  of training data are used. Instead of training the model on BirdSet's Xeno-Canto
  cut or iNaturalist's iNatSound dataset, the authors scrape the data anew.
- While the model achieves state-of-the-art results, the improvements over
  previous methods are relatively small.
- The comparison on the BEANS benchmark lacks strong baselines as identified in
  [1]. BirdMAE and the encoder of NatureLM-audio could also be tested with LP
  and PP.
- Tables in the appendix are hard to read, as the best and second-best results are
  not highlighted.


[1] Foundation Models for Bioacoustics -- a Comparative Review
https://www.arxiv.org/abs/2508.01277

**Questions:**

- How many seeds were used to validate the results? What are the standard
  deviations?
- What is the reasoning behind selecting 6 s and 5 s windows (lines 130–131)?
- Why four prototypes per class (line 193)?
- How much faster is training with softmax compared to a binary-sigmoid objective (line 207)?
  Can you share some details?
- What is your rationale behind setting the number of training steps? Did you
  use early stopping?
- "General purpose audio benchmarks such as HEAR are still dominated by supervised and semi-supverised models..." (l.424f.) What is your opinion on the recent results on the AudioSet benchmarks, where models like BEATs [2] or SSLAM[3] outperforming SL models?

[2] BEATs: Audio Pre-Training with Acoustic Tokenizers https://arxiv.org/abs/2212.09058
[3] SSLAM: Enhancing Self-Supervised Models with Audio Mixtures for Polyphonic Soundscapes https://arxiv.org/abs/2506.12222

---

> ### Author Response · Authors · 2025-11-26
>
> We thank Reviewer TSZg for their feedback.
>
> > Ablation studies
>
> As noted in our high-level feedback, we have added several ablations to the appendix.
>
> We hope this addresses the Reviewer's concern with the difficulty of assessing the importance of each individual change. In particular, we find that while ablating any individual design choice has some detrimental impact on performance, the biggest performance drop was observed when restricting the training data to Xeno-Canto only.
>
> > Question: does the "supervision remains dominant in bioacoustics" conclusion still hold if Perch 2.0 is trained on the same data as BirdMAE?
>
> The training paradigms of BirdMAE and Perch 2.0 are sufficiently different that it is difficult to hypothesize which model would perform best given the exact same dataset. Moreover, the training pipeline of BirdMAE is specifically geared towards self-supervised learning (e.g., they limit the number of windows per species to reduce class imbalance, which is a particular concern for self-supervised methods). This makes it even harder to fairly compare the two models without a more in-depth study.
>
> However, we don’t believe that this takes away from the overall “bittern lesson” that we are presenting: rather than focusing on more complex learning algorithms, we can make substantial progress in bioacoustics by staying with a simple supervised learning paradigm and collecting more annotated training data.
>
> > No fixed sets of training data are used, and the data is scraped anew.
>
> The reviewer's point on reproducibility is well-taken, and we commit to publishing a CSV of pointers to all the audio files used for training along with the camera-ready submission.
>
> We note however that substantiating the claim that additional supervised training data is a major driver of performance improvement necessarily requires departing from established training sets.
>
> > Relatively small improvements over previous methods.
>
> We do not claim that the improvements are substantial: Our contribution is not to provide a model which outperforms existing approaches by a large margin, but rather to report on our observation that well-tuned supervised models using strong data augmentations and auxiliary training objectives can perform comparably to or even better than models that have one or two orders of magnitude more parameters, use sophisticated self-supervised learning algorithms, or employ large language models.
>
> > BEANS comparison lacks strong baselines, like BirdMAE and NatureLM-audio's encoder paired with LP and PP.
>
> We compare against the evaluation metrics provided by prior works. BirdMAE only reports results on BirdSet, and NatureLM-audio does not report performance metrics for its encoder specifically.
>
> We thank the Reviewer for pointing out Schwinger et al.'s contemporaneous work [1], which reports top-1 accuracies on BEANS' classification tasks for BEATs NLM and BirdMAE. Please refer to our high-level comments for a detailed comparison with the results reported in [1].
>
> In light of those results, we stand by our main claim: Well-tuned supervised models using strong data augmentations and auxiliary training objectives can perform comparably to or even better than models that have one or two orders of magnitude more parameters, use sophisticated self-supervised learning algorithms, or employ large language models.
>
> > Tables in the appendix are hard to read, as the best and second-best results are not highlighted.
>
> We thank the reviewer for bringing this up; we highlighted the best and second-best results in the latest revision.
>
> > Question: how many seeds were used to validate the results, and what are the standard deviations?
>
> Regretfully we did not have time to rerun our experiments in order to calculate standard deviations on the test sets.
>
> However, note that to make the exploration of the hyperparameter space feasible we used Vizier, a black-box hyperparameter optimization tool. Each model trained by Vizier used a different random seed. Moreover, if a region of hyperparameter space has particularly high variance, Vizier’s Gaussian Process Bandits algorithm is unlikely to explore this space further. Hence this hyperparameter optimization approach should naturally avoid overfitting to a random seed.
>
> Also note that we would not suggest that people regularly retrain (foundational) bioacoustics models like Perch 2.0 from scratch (which would require robustness to hyperparameter choices). Instead, we have released the model weights so that people can use the species predictions, train probes, or fine-tune the model.

---

> ### Author Response · Authors · 2025-11-26
>
> > Question: why four prototypes per class?
>
> In their AudioProtoPNet paper, Heinrich et al. find that increasing the number of prototypes beyond 5 does not increase performance further. We used 4 simply because as computer scientists we have the habit of using powers of two wherever possible.
>
> > Question: how much faster is training with softmax compared to a binary-sigmoid objective?
>
> We found experimentally that using a softmax loss meant that the model could get to ~90% of its final performance at about twice the speed, which made exploring the hyperparameter space easier. When training for longer the models using sigmoid binary cross-entropy would eventually achieve the same performance. Hence, the choice here was simply made to shorten the feedback loop during development; we don’t believe that using either loss materially affects the final results.
>
> > Question: what is the rationale behind setting the number of training steps, and was early stopping used?
>
> We used early stopping and chose a number of steps that was roughly twice as long as the number of steps that our models generally took to converge to ensure that we were not underfitting our models.
>
> > Opinion on the recent results on the AudioSet benchmarks, where models like BEATs or SSLAM outperform SL models.
>
> We are excited to see whether models like BEATs and SSLAM will have an impact in bioacoustics. In our experience–which is corroborated by work such as the HEAR benchmark–general sound event detection and bioacoustics are sufficiently different that conclusions in the former don't necessarily translate into the latter.

---

### Official Review · Reviewer_Xkp3 · 2025-11-01

**Soundness:** 2
**Presentation:** 3
**Contribution:** 3
**Rating:** 6
**Confidence:** 4

**Summary:**

Perch 2.0 is a compact bioacoustic model trained on ~14 000 species and event classes to produce general-purpose audio embeddings. It keeps the backbone of Perch 1.0 but adds three elements: a prototype-based species head, a linear head trained through self-distillation from the prototype outputs, and a self-supervised source-prediction task. Strong data augmentations (generalized mixup, random windows) handle label noise.
Across 19 benchmarks, Perch 2.0 claims to achieve state-of-the-art transfer and few-shot performance while remaining small enough for field deployment. The authors argue that careful supervised training with auxiliary objectives can match or surpass much larger self-supervised audio models.

**Strengths:**

- The paper combines supervised training, prototype-based distillation, and auxiliary objectives in a clear, effective design.

- Experiments cover a wide range of benchmarks and are technically solid.

- The model transfers well across domains while staying compact and efficient.
- Strong performance under linear probing shows the embeddings are general and practical to use.

- The model architecture is optimized to be employed in real-world systems, so as to be as light as possible.

**Weaknesses:**

- Unclear contribution of components to performance: The paper introduces several methodological components (e.g., multi-source mixup, self-distillation, and an auxiliary source-prediction loss). However, their individual contributions are not clearly isolated, as the paper does not provide controlled ablation studies. In particular, the role of the windowing strategy and the handling of label noise across heterogeneous sources remains insufficiently explained. This raises concerns regarding reproducibility and makes it hard to disentangle how components are contributing to the reported performance gains.

- Missing engagement with contrasting literature: The paper does not currently cite or discuss [1], which provides a systematic comparative analysis of multiple models, including Perch 1.0, on the same BEANS and BirdSet benchmarks. That work concludes that self-supervised models such as BirdMAE generally outperform supervised approaches. Including and positioning the present findings in relation to [1] would strengthen the contribution by clarifying how this work complements existing results.

- Dataset construction lacks some detail: the paper states that classes were manually mapped, but the procedure and criteria for this mapping are not fully described. More detail on how label merging/normalization was performed (e.g., handling synonyms) would help ensure reproducibility and interpret the consistency of supervision across sources.

- Overall this is a good paper, with strong results and clear relevance to the field. The main gaps are the lack of ablation studies, some missing information on dataset construction and limited positioning relative to recent literature: addressing these points would substantially strengthen the clarity and credibility of the contribution. Additionally, a plan for code release is not mentioned but would further support reproducibility. With these additions, I would be in favor of a strong acceptance of the paper.

[1] Schwinger, R., Zadeh, P. V., Rauch, L., Kurz, M., Hauschild, T., Lapp, S., & Tomforde, S. (2025). Foundation Models for Bioacoustics--a Comparative Review. arXiv preprint arXiv:2508.01277.

**Questions:**

1. I understand that running full pretraining ablations may be computationally prohibitive within the rebuttal timeline, but additional clarification would be helpful. Could the authors share thoughts on the design choices and how they are expected to drive performance?

2. The fixed 16 kHz input bandwidth excludes ultrasonic frequencies, yet the paper reports strong performance on marine mammals and bats. How does the model handle higher-frequency taxa such as dolphin or bat?

3. Could you provide clarification or additional results showing how each architectural component affects performance? Specifically, how does windowing and noise in labels affect the final outcomes?

4. The use of random vs. energy-peak 5 s windows is underexplained (section 2.1). What is the proportion of data processed by each method? What is their comparative effect on performance?

5. Can you provide more details on how you handle normalization/merging of labels across different datasets ( Xeno-Canto, iNaturalist and the Tierstimmenarchiv)?

6. Will the code be released upon acceptance?

---

> ### Author Response · Authors · 2025-11-26
>
> We thank Reviewer Xkp3 for their feedback.
>
> > Ablations and discussion of design choices
>
> As noted in our high-level feedback, we have added several ablations to the appendix.
>
> As noted in the revised text, these demonstrate that the major driver of performance is the availability of additional supervised data since the original release of the Perch model. Modifying the Perch 2.0 Phase I model by removing index prediction, reducing mixup, and restricting to Xeno-Canto data all lead to somewhat reduced performance relative to the final model.
>
> We have not explored window size in this study. The 5-second window matches the previous Perch model, and was chosen as a trade-off between the model's context size and locality of windowed predictions.
>
> We agree that label noise is a topic very worthy of exploration, but it is beyond the scope of this work, and deserves a dedicated study.
>
> > Discussion of Schwinger et al. [1]
>
> Please refer to our high-level comments for a detailed comparison of the results in Schwinger et al. and Perch 2.0’s performance.
>
> We would like to make the passing comment that [1] is a pre-print (i.e., not peer reviewed) and that the [reviewer guide](https://iclr.cc/Conferences/2026/ReviewerGuide#:~:text=Are%20authors%20expected%20to%20cite%20and%20compare%20with%20very%20recent%20work%3F) for ICLR 2026 establishes that work up to two months prior to the submission deadline is considered contemporaneous (and hence does not need to be compared against).
>
> > Dataset construction details
>
> We scraped recordings as follows:
>
> * Xeno-Canto was scraped using the official API
> * iNaturalist was scraped using the “Research grade” recordings listed [on GBIF](https://www.gbif.org/dataset/50c9509d-22c7-4a22-a47d-8c48425ef4a7)
> * Tierstimmenarchiv was manually scraped
>
> After downloading the audio files, we mapped the species labels of different datasets to the taxonomy used by iNaturalist (since this taxonomy encompasses the widest number of species and is the most up to date). The mapping was a partly manual process that roughly proceeded as follows:
>
> 1. If the scientific species name exists in the iNaturalist taxonomy, use the identity mapping
> 2. If the species is a bird then look up the species in the Avibase database to check if there is a mapping to the Clements 2024 taxonomy
> 3. If the species was split into multiple species, arbitrarily pick the first one
> 4. If the species has a common name that matches the common name on iNaturalist, map to the corresponding species
> Otherwise, resolve manually (using Google, Wikipedia, etc.); this had to be done for a several dozen species (most cases involved spelling differences or data entry errors)
>
> The result of our manual labor is encapsulated in a Python library which we have already open-sourced. The library provides the exact mapping between classes that was used to develop our model. We cannot link to it to avoid breaking anonymity, but we commit to doing so in the camera-ready revision.
>
> > Code release plan
>
> An open-weights Perch 2.0 model checkpoint is already distributed on Kaggle. We cannot provide the link at the moment to avoid breaking anonymity, but we commit to do so in the camera-ready revision. Our training code is intertwined with a variety of libraries that have not been publicly released, which makes it difficult for us to provide a fully-functioning training pipeline, but we commit to releasing reference implementations for the important parts of our model jointly with the camera-ready revision (e.g., generalized mixup implementation, ProtoPNet head, and so on).

---

> ### Author Response · Authors · 2025-11-26
>
> > How does the model handle higher-frequency taxa such as dolphin or bat?
>
> For the BEANS datasets we simply read all of the data as if it has a sampling rate of 32 kHz. Hence recordings are pitch shifted depending on the input sampling rate; for the Egyptian fruit bats this meant that time was slowed down by a factor 7.81. Note that the cetacean vocalizations in the BEANS dataset do not contain ultrasonic sounds (the Watkins marine mammal dataset is sampled at 44.1 kHz).
>
> For the transfer tasks from Ghani the bat vocalizations were already pitch-shifted. The exact process is not detailed in the Ghani paper beyond “pitch shifting via sample rate conversion”, but our educated guess is that the audio was upsampled to 441kHz and then slowed down by a factor of 10 (by reading 441 kHz audio as if it was 44.1 kHz). Since bat calls generally range from 10 kHz to 150 kHz, this would bring all bat calls into the audible range of 1 to 15 kHz (our mel-spectrograms go as high as 16 kHz).
>
> We added this clarification to the submission.
>
> > What is the proportion of data processed by random and energy-peak selection methods, and what is their comparative effect on performance?
>
> We either select random windows or select windows based on the energy peaks. We never combine the two selection methods in a single experiment. For the first phase (Perch 2.0 - Phase I) we use random window selection and for the second (self-distillation) phase we experiment with either methods (Perch 2.0 - Peak-select, Perch 2.0 - Random). Table 3 in the main text compares performance in all three cases. For the second phase, selecting based on the energy peaks has a negligible effect on performance, save for the top-1 accuracy metric on BirdSet, where it degrades performance when compared with random window selection.
>
> As for the number of selected windows in the training data, when we select windows based on the energy peaks we select an average of 4.15 windows per recording: a total of 898,930 recordings resulted in 3,732,225 6-second segments.

---

### Author Response · Authors · 2025-11-26
**Rebuttal**

We thank all the reviewers for their time and feedback. We will respond to each reviewer individually, but would also like to provide some high level comments. We posted a revision with changes color-coded in blue for ease of viewing. In short, we:

* moved Figure 2 caption below the diagram;
* added a reference to Figure 2 in the main text;
* updated BirdSet, BirdMAE, and CLAR references to cite the conference versions;
* added an explanation in the main text on how bat vocalizations are handled in BEANS evaluations of Perch 2.0;
* added an Ablations section in the Appendix, along with a reference to it in the main text;
* added Schwinger et al.'s BEATs NLM and BirdMAE results on BEANS classification tasks in the main results table and updated numbers reported in that column so as to make them comparable; and
* highlighted the best and second-best results in the tables in the main text and the Appendix.

# References

[1] Schwinger, R., Zadeh, P. V., Rauch, L., Kurz, M., Hauschild, T., Lapp, S., & Tomforde, S. (2025). Foundation Models for Bioacoustics--a Comparative Review. arXiv preprint arXiv:2508.01277.

---

> ### Author Response · Authors · 2025-11-26
> **Ablations**
>
> In response to the reviews we have added an appendix with ablations of the most important model design choices. We want to contextualize these ablations by making two observations:
>
> Firstly, removing or changing a single model design choice without fully re-tuning the other hyperparameters has limitations. For example, if we remove one feature that acts as a regularizer (like mixup) but not increase regularization elsewhere (by increasing the learning rate or dropout) the model might be under-regularized. However, the alternative (fully re-tuning the model for each ablation) is computationally prohibitive.
>
> Secondly, our paper’s central claim is not that any specific model design choice (such as generalized mixup and index prediction) is of particular scientific interest because of their novelty or large impact on model performance. In fact, in some ways we are arguing the opposite: Our final model is remarkably small and relatively simple, but still outperforms models that have one or two orders of magnitude more parameters, use sophisticated self-supervised learning algorithms, or employ large language models. This paper explores why this might be the case in the hope that it helps inform future research directions (e.g., we believe this could motivate research on label cleaning and learning from noisy labels, rather than advanced self-supervised models).
>
> That is not to say that we believe our work was trivial: we spent significant time and effort on assembling datasets, harmonizing taxonomies, and carefully exploring the space of data augmentations, architectures, and training paradigms. This work goes against the general trend in bioacoustics (and machine learning more broadly) to develop complex self-supervised models, and as such we strongly believe our findings are a valuable addition to the conversation about the state of bioacoustics modelling and future research directions.

---

> ### Author Response · Authors · 2025-11-26
> **Baseline comparisons**
>
> Several reviewers highlighted the recent study by Schwinger et al. (2025) [1] which concluded that BirdMAE and BEATS NLM (two self-supervised models) were the best performing models on BirdSet and BEANS respectively. We would like to highlight why we do not think these results impact our claims that Perch 2.0 achieves state-of-the-art results and that a small, well-tuned supervised model can still outperform far larger and more complex models.
>
> Firstly, we must ensure the same metrics are used (e.g., [1] does not evaluate on ESC-50 and Speech Commands). For the remaining BEANS classification tasks, looking at Table 10 in [1] shows that the best-performing model (BEATs NLM with attentive probing) achieves 83.65% top-1 accuracy on average, whereas Perch 2.0 - Random achieves 83.6% top-1 accuracy on average with prototypical probing. For BirdSet, looking at Table 7 in [1] shows that the best-performing model (BirdMAE with attentive probing) achieves 86.54 AUROC on average, whereas Perch 2.0 - Random achieves 90.8 AUROC on average.
>
> **In other words, Perch 2.0 performs comparably to the best self-supervised models on BEANS' classification tasks, and it outperforms the best self-supervised models on BirdSet.**
>
> However, there are two ways in which we cannot make an apples-to-apples comparison with the numbers reported in [1]: Firstly, Perch 2.0’s scores on BirdSet are achieved without any learning on the BirdSet training data (the “Restricted” setting in [1]). Secondly, on the BEANS tasks, attentive probing gives BEATS NLM (and BirdMAE) a much larger number of trainable parameters. Attentive probing has $2D^2 + (C + 1)D + C$ parameters compared to prototypical probing’s $C(J(D + 1) + 1)$. As an example, on the bats subtask of BEANS (where $C = 10$) this gives BEATS NLM (which has $D = 768$) a total of 1,188,106 learnable parameters. Our prototypical probing (with $J = 4$) only has 61,490 parameters (19 times fewer). Both of these differences arguably put Perch 2.0 at a disadvantage.
>
> As we explain in our manuscript, we believe that successful downstream deployment of bioacoustics models requires the ability to adapt with only one or a handful of labeled examples (e.g., clustering, nearest neighbor searches or agile modeling setups). Hence real-world utility is likely best measured using simple read out layers (such as linear or prototypical probes) that are less prone to overfitting on small amounts of data.

---

### Note · Program_Chairs · 2026-01-22
**Submission Desk Rejected by Program Chairs**

The use of the possessive "our" to refer to prior work violates double blind and the submission must be desk rejected.